# Reproductive seasonality in the Baka Pygmies, environmental factors and climatic changes

**Laura Piqué-Fandiño[1], Sandrine Gallois[2], Samuel Pavard[1], Fernando V. Ramirez Rozzi** [1,3]*

**1** Eco-anthropologie (EA), Muséum national d'Histoire naturelle, CNRS, Université de Paris, Musée de l'Homme, Paris, France, **2** Universitat Oberta de Catalunya, Barcelona, Spain, **3** EA 2496, Faculté de Chirurgie Dentaire, Université de Paris, Montrouge, France

* fernando.ramirez-rozzi@mnhn.fr

## Abstract

Reproductive seasonality is a phenomenon common to human and animal populations and driven by, among others, climatic variables. Given the currently changing climate and its impacts on both the environment and human lives, the question arises of its potential effects on reproductive seasonality. Few studies have specifically explored the seasonality of reproduction among hunter-gatherers and anyone investigated how current climate change might affect this phenomenon. In this study we addressed reproductive seasonality in the Baka Pygmy living in African rain forests. Since reproductive seasonality can be linked to weather patterns, we explore this possibility. However, climatic variables driving weather patterns have changed over the years, so we assessed whether this has influenced the Baka reproductive pattern. Based on 34 years of written birth records and oral questionnaires from 13 years of systematic fieldwork, we observed a bimodal birth pattern with two birth peaks at 6-month intervals. Our results demonstrate that precipitation at conception or at birth potentially has effects, respectively negative and positive on the monthly number of births; and temperature has a role in controlling other variables that do affect the reproductive pattern. Changing weather patterns appear to be affecting the reproductive seasonality in the Baka, suggesting that attention needs to be given to the influence of global climate change on forager societies.

## Introduction

Studying human reproductive seasonality, i.e. the seasonality of conception and birth, is of importance because birth months have significant consequences for many demographic aspects such as infant mortality [e.g. 1, 2], maternal performance [3], adult height [4], lifespan [5, 6] and susceptibility to late-onset diseases [e.g. 7–9]. This is a classic issue in demography [e.g. 10, 11], whose significance has been demonstrated in the vast majority of populations from the broad national scale [e.g. 12] to the local scale [13], but with considerable differences in pattern, magnitude and timing [e.g. 12].

**Data Availability Statement:** All relevant data are within the paper and its Supporting information files.

**Funding:** The author(s) received no specific funding for this work.

**Competing interests:** The authors have declared that no competing interests exist.

Seasonal reproductive patterns are observed not only in human populations, but also in other mammal species such as rodents and artiodactyls [14], wild dogs [15] and primates [16–18]. Although great apes do not have restricted birth seasons, they do show seasonal distributions in the frequency of births [19]. For instance, in chimpanzees, conceptions tend to occur during periods of increased food supply [16] while in free-ranging orangutan populations, it is related to the mast fruiting season and to body condition [20].

Energetic determinants play an important role in reproductive seasonality. Because pregnancy is a process that demands high energy expenditure, conceptions must be concentrated in periods when the energy balance is positive [21]. Thus, a good nutritional status has been indicated as a key factor in higher conception rates [12, 21, 22]. While food supply and body condition seem to be the cause of seasonality in other primate species, in humans, the biological environmental and socio-cultural determinants of reproductive seasonality are highly variable among populations. Beyond the nutritional status, other determinants such as behavioural (related to socio-cultural and economic context) and climatic factors might play an important role in shaping reproductive seasonality [19]. Firstly, seasonal socio-cultural aspects such as marriages, holidays, periods of work and religious celebrations [e.g. 23–25] shape the distribution of births over a year in several populations [26]. Secondly, environmental or climatic determinants such as temperature [e.g. 27, 28] and rainfall [e.g. 13, 29–31] also shape reproductive seasonality in several societies. For instance, rates of conception are lower in extreme temperatures [e.g. 27, 28], which may be due to a lower frequency of intercourse [e.g. 30, 32] or to biological factors affecting conception [e.g. deterioration of semen quality during warm months, 33]. Rainfall has proved to shape the reproductive seasonality through differential food availability [13, 21, 30] or increased incidence of infectious diseases, such as malaria, which may in turn depress fecundability [29, 31]. Additionally, rainfall and temperature are associated with many human subsistence activities that can modulate the frequency of intercourse as well as the population's perception of the auspiciousness of having a child. As regards the latter factor, weather conditions at the time of birth, not conception, may therefore also influence birth seasonality [34], as parents might choose to give birth in periods with better climatic conditions that improve food availability and reduce the likelihood of diseases. Other environmental characteristics such as latitude are seen as determinants of the timing of the human birth peak, with peaks coming earlier in the year at locations further from the equator [35].

Even though reproductive seasonality is a classic topic of investigation in anthropological and historical demography [e.g. 6, 11, 26, 30], only few studies have investigated the presence and causes of reproductive seasonality in hunter-gatherer societies [13, 36, 37]. These societies depend on a very close relationship with the environment for their culture and livelihood (including food resources and medicine), and can thus be expected to be more directly influenced by climatic variables (i.e. rainfall and temperature). The small amount of literature on hunter-gatherers might be explained by a lack of access to accurate data. Unlike studies in agricultural and industrialized societies, studies of demographic aspects such as birth seasonality in hunter-gatherers raise particular challenges in terms of data quality and quantity. The lack of chronology and the absence of written records introduce a bias whose magnitude is unknown in most cases. Among the few studies exploring reproductive seasonality in hunter-gatherers, Bailey et al. [13] did not observe any birth pattern in the Efe Pygmies from the Ituri forest. Among the San from the Kalahari, Wilmsen [36, 38] found that nutritional status (mediated by environmental variables such more precipitation and by cultural mechanisms) is the main factor influencing the birth seasonality. Additionally Hill and Hurtado [37], in their study on the Ache from Paraguay, did not relate birth seasonality directly to nutritional status but rather to the energy balance, since they found that fewer conceptions correlated with lower

body weights. Therefore, the association of environmental variables, and more specifically climatic variables, with reproductive seasonality among hunter-gatherers seems diverse.

The fact that climatic variables might relate to reproductive patterns and reproductive seasonality in human populations raises another topical issue. The cascading effects of climate change in these last decades are not only affecting the physical and biological components of the planet (such as sea level, ice retreat, freshwater availability and biodiversity loss), but also the means of subsistence, health and well-being of human populations [39, 40]. For instance, Philibert et al. [31] have observed that recent climatic changes have influenced birth seasonality of rural societies in Mali. Indigenous peoples and local communities, such as hunter-gatherers, are indeed considered to be the populations the most vulnerable to climate change because of their overall marginalization, lack of land tenure and poor access to health services and justice systems [41]. Therefore, considering the importance of reproductive seasonality for demographic and health issues, it is worth investigating whether climate change might be impacting the reproductive pattern among these populations.

The so-called Pygmy populations are semi-nomadic hunter-gatherers living in the equatorial rain forests of Africa. They have an independent evolutionary history that split from non-Pygmy populations around 60000BP [42]. Two clusters of Pygmy groups then split out around 2800 BP [43]: one in West Africa (the Twa, Kola, Bongo, Koya, Aka and Baka groups) and the other in East Africa (the Aka, Sua, Mbuti/Efe groups and the BaTembo). African Pygmies grow to an average adult stature of <155cm [44]. The genetic bases of this very specific phenotype are polygenic. They include genes from the growth hormone axis as well as genomic and gene regions that relate to the traits themselves [45–47]. Pygmy morphology has long been interpreted as an adaptation to forest life [44, 45] in an extremely humid and dense environment [48].

Among these different Pygmy groups, the Baka have a geographical distribution comprising south-eastern Cameroun, north-eastern Gabon and north-western Congo. Based on genetic evidence and despite their semi-nomadic lifestyle, the pattern of Baka dispersal is highly localized, with a maximum parent-offspring dispersal of approximately 60km [49]. The climate in the Guineo-Congolese evergreen forest [50–52] where they became established is typical of inland areas near the equatorial zone, with year-round moderate to high temperatures (monthly mean ≈23ºC) and seasonal rainfall with two wet and two dry seasons annually and average monthly precipitation of <1540mm [53, 54]. The rainy seasons are characterized by short periods of torrential rainfall [55]. Depending on the specific geographic area, the months and duration of the seasons may vary slightly by more or less one month.

The Baka present a subsistence economy based on foraging activities, hunting, gathering and fishing, combined with a recently introduced small-scale farming [56]. The habitat where the Baka live provides them with food all year around, from both wild meat, plants and other edibles, and agricultural crops. The Baka do not present nutritional deficiencies [57] and no episode of starvation or famine has been recorded [58]. However, there are seasons when specific foods are scarcer or more abundant [59, 60]. The fluctuations in the availability of wild food resources and in agricultural work imply that Baka subsistence activities vary according to season [56]. Overall, agricultural tasks are conducted all year long, but more intense work is needed when oppening plots, in both dry seasons, and when planting groundnuts (around April). Foraging patterns also vary with some important periods of more intense activity. For instance, gathering wild or "bush" mangoes (Irvingiaceae family), mostly from June to August [61], is a fundamental activity and a key period in the Baka calendar. During this time, all adult men and women, as well as children, spend most of their time gathering these fruits [59, 61]. The intensity of gathering is such that bush mango reserves are generally depleted by the end of July or the beginning of August (FRR, personal observation). Hunting activities, performed

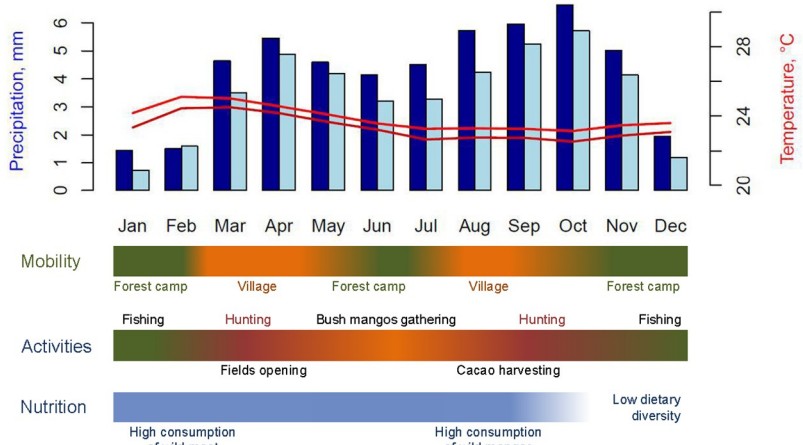

**Fig 1. Weather variables and economic activities.** Daily mean precipitation (mm) and temperature (Cº) by month for the years 1979–2002 (dark blue bars and dark red line) and 2003–2018 (light blue bars and light red line), and corresponding timeline of predominant economic activities. It must be stressed that, due to the bimodal distribution of precipitation through the year, precipitation in a given month is strongly inversely correlated with precipitation 9 months later (cor = -32%, p = 8.5e-11). Temperature in the 9-month intervals is less correlated (cor = 9%, p = 0.06). Data from ERA5 [64]. Mobility shows the place where the Baka live after the season [56]. Activities present the most relevant subsistence activities by season [61, 65]. Nutrition presents the most relevant insights regarding Baka food behaviours for our study [61, 66].

mainly by the men, are more intense during both rainy seasons (September to December and March to June). Fishing, mainly done by women, is more frequent during the dry seasons, from December to March and from July to August. Moreover, even if women and men invest more time in fishing and hunting respectively, the evidence shows that women also spend time on hunting and men on fishing [62, 63].

Precipitation data from the study site (S1 Fig and S1 Table) show a bimodal pattern through the year with two rainy seasons (March-May and August-November) and two dry seasons (more intense in December-February and less intense in June-August) at 6-monthly intervals. Temperature is moderate to high all year round (daily mean of ~23ºC) with no wide yearly oscillation (S1 Fig and S2 Table). Weather conditions govern the Baka economic calendar (Fig 1): hunting is especially important during rainy seasons whereas fishing and bush mangoes gathering are intense during dry seasons [61, 62]. Subsistence activities clearly follow climatic fluctuations (Fig 1) so it can be difficult to distinguish whether a specific outcome is influenced by subsistence activities or by the climate.

This study therefore has three main aims. The first is to assess the presence and pattern of reproductive seasonality in the Baka Pygmies who live in the African rainforest. Since previous studies of the Ache in the rainforests of eastern Paraguay have demonstrated birth seasonality [29], we expect the Baka to show reproductive seasonality as well. Studies on reproductive seasonality have found birth peak amplitudes ranging from 5 to 65% [12, 23, 27, 29]. Recent studies on growth and life history variables in the Baka Pygmies in Cameroon [58, 67] have shown a modal interbirth interval (IBI) of 2.5 years. Since the IBI is 2.5 years, successive siblings are born at two different times in the year. If a peak marks a seasonal birth pattern in the Baka, we would expect a second peak six months later and thus a two-peak pattern of birth seasonality. Two-peak reproductive patterns are common and present in several other societies [23, 30, 31]. Secondly, we aim to explore whether climatic variables, such as temperature and rainfall, might be associated with the reproductive pattern in the Baka. Several previous studies have

found a link between weather variables and reproductive seasonality, in particular conception in periods of positive energy balance due to greater food availability and quality and thus better ovarian function [e.g. 16, 20, 27, 29, 30, 37]. If this is true, we would expect a conception peak in the Baka Pygmies in the dry season when the overall availability of wild food is greater [i.e. 59, 68]. Thirdly, other studies [22, 29, 31] state spontaneous abortions or diseases affecting conceptions are more likely to happen during wetter seasons, so we would expect a conception peak during dry season. Fourthly, social factors might also play a role. Subsistence activities vary according to the seasons, and Baka men might spend days and weeks further from their families, engaged in hunting expeditions or agricultural related activities during the rainy seasons [56]. Also, as other studies have outlined high workload periods might decrease conception rates [21]. Thus, we expect conception rate to be lower during rainy seasons when subsistence activities away from the households are longer and the workload is higher. According to our previous hypotheses, we would expect a depression of fecundability during rainy seasons and a conception peak during dry seasons due to greater food availability, less disease incidence and subsistence activities closer to the households. Finally, if climatic and reproductive patterns are related, our third aim is to investigate the relationship between global climate change and local reproductive patterns. Because temperatures are increasing worldwide and rainfall patterns are changing in different parts of the planet due to climate change, we expect to see variations in both variables over the last 30 years, with an effect on reproductive seasonality [e.g. 11, 27, 69]. This study is based on 13 years of systematic and regular fieldwork and on 34 years of birth records, the largest number of such records ever obtained (n = 1803) for a hunter-gatherer population ([13] n = 147, [36] n = 221).

## Materials and methods

### Materials

**The Baka Pygmies.** The study took place in Moange Le Bosquet (3º05'07"N 14º03'45"E), a Baka Pygmy village in south-eastern Cameroon. Founded in 1973 around a Catholic mission, Le Bosquet has a medical centre run by nuns with medical training whose community was established when the village was founded, to provide health care and keep records of births in the Baka community. The Baka from Le Bosquet share their life between the village itself and camps in the middle of the forest, about 15 km away. The births of Baka children are registered in the centre's records even when the families are spending time in their forest camps, since travel between the camps and the village is frequent. Birth records with both parents identified were available to us from 1980 to 1982 and from 1988 until now. According to several censuses carried out by the nuns and FRR over the last 13 years, the number of Baka individuals in Le Bosquet has remained more or less constant since 1973 at around 800 individuals, divided into approximately 230 households. Thanks to the nuns' and our censuses, we know that approximately half of the households register births in the medical centre. This subpopulation in Moange Le Bosquet makes up the subject of our study. It comprises 406 households, which is higher than the total number of households in Le Bosquet in any given year because the birth records over 13 years include many households who were living in Le Bosquet but moved out after some years. The constant number of inhabitants in Le Bosquet since its foundation implies that the number of families moving in and out is more or less the same.

Prior informed consent was obtained from all participants and the study was carried out with the approval of the chief of the Baka community. All methods are non-invasive and were carried out in accordance with relevant guidelines and regulations. This study has obtained approval from the Centre National de la Recherche Scientifique, the Agence National de la Recherche (France) and from the Institut de Recherche et Développement. It was carried out

under an international agreement between the Institut de Recherche pour le Développement (IRD) and the Ministry of Scientific Research and Technology of Cameroon. It obtained the approval of the National Committee of Ethics for Research for Human Health of Cameroon (2018/06/1049/CE/CNERSH/SP).

**Data.** The data comprise a total of 1083 records of births over 34 years (from 1980 to 1982 and from 1988 to 2018) (Table 1). These records include the names of both parents and the exact date of birth (day, month and year; only 14 records lack the exact day). The number of births per year ranges from 16 to 55 with an average of 31.85 (SD = 8.4). The number of women of child-bearing age (15 to 55) present in the village every year is known from 2007 to

**Table 1. Number of births by month and by number of women present in Le Bosquet.**

| Year | Jan | Feb | Mar | Apr | May | Jun | Jul | Aug | Sep | Oct | Nov | Dec | Total | Women |
|------|-----|-----|-----|-----|-----|-----|-----|-----|-----|-----|-----|-----|-------|-------|
| | | | | | | Number of birth | | | | | | | | Women |
| 1980 | 3 | 3 | 1 | 2 | 2 | 1 | 0 | 1 | 3 | 1 | 0 | 3 | 20 | |
| 1981 | 1 | 1 | 1 | 0 | 0 | 1 | 0 | 2 | 3 | 2 | 2 | 5 | 18 | |
| 1982 | 0 | 3 | 1 | 4 | 1 | 2 | 5 | 1 | 2 | 3 | 1 | 2 | 25 | |
| 1988 | 0 | 2 | 0 | 2 | 4 | 1 | 0 | 3 | 2 | 3 | 3 | 4 | 24 | |
| 1989 | 2 | 3 | 3 | 3 | 4 | 1 | 2 | 2 | 0 | 7 | 2 | 1 | 30 | |
| 1990 | 2 | 1 | 4 | 7 | 4 | 2 | 3 | 4 | 1 | 2 | 1 | 6 | 37 | |
| 1991 | 2 | 1 | 5 | 3 | 3 | 4 | 5 | 4 | 4 | 1 | 4 | 2 | 38 | |
| 1992 | 5 | 3 | 1 | 5 | 3 | 0 | 5 | 4 | 3 | 3 | 3 | 1 | 36 | |
| 1993 | 1 | 1 | 4 | 2 | 1 | 4 | 4 | 6 | 4 | 2 | 4 | 3 | 36 | |
| 1994 | 3 | 1 | 2 | 2 | 5 | 1 | 3 | 3 | 3 | 2 | 5 | 2 | 32 | |
| 1995 | 2 | 6 | 5 | 2 | 1 | 0 | 1 | 0 | 5 | 1 | 3 | 5 | 31 | |
| 1996 | 3 | 4 | 1 | 3 | 6 | 3 | 5 | 3 | 3 | 8 | 2 | 1 | 42 | |
| 1997 | 1 | 3 | 3 | 4 | 1 | 0 | 5 | 1 | 0 | 3 | 2 | 2 | 25 | |
| 1998 | 2 | 1 | 3 | 4 | 3 | 2 | 1 | 4 | 1 | 2 | 3 | 1 | 27 | |
| 1999 | 2 | 3 | 3 | 6 | 3 | 1 | 1 | 5 | 3 | 3 | 5 | 4 | 39 | |
| 2000 | 6 | 2 | 2 | 4 | 4 | 2 | 3 | 3 | 0 | 6 | 2 | 4 | 38 | |
| 2001 | 4 | 6 | 2 | 0 | 6 | 6 | 0 | 7 | 4 | 3 | 0 | 3 | 41 | |
| 2002 | 3 | 1 | 3 | 1 | 3 | 2 | 2 | 2 | 5 | 4 | 8 | 1 | 35 | |
| 2003 | 1 | 4 | 0 | 1 | 2 | 4 | 1 | 3 | 3 | 3 | 2 | 6 | 30 | |
| 2004 | 2 | 6 | 3 | 3 | 8 | 2 | 1 | 0 | 2 | 6 | 2 | 4 | 39 | |
| 2005 | 5 | 3 | 5 | 3 | 5 | 5 | 6 | 4 | 4 | 8 | 4 | 3 | 55 | |
| 2006 | 0 | 5 | 1 | 3 | 3 | 2 | 0 | 0 | 2 | 5 | 3 | 1 | 25 | |
| 2007 | 1 | 2 | 2 | 1 | 3 | 0 | 1 | 2 | 1 | 3 | 1 | 1 | 18 | 88 |
| 2008 | 3 | 3 | 2 | 6 | 4 | 7 | 1 | 2 | 4 | 0 | 4 | 2 | 38 | 109 |
| 2009 | 4 | 2 | 3 | 5 | 3 | 3 | 8 | 1 | 3 | 3 | 1 | 2 | 38 | 155 |
| 2010 | 5 | 4 | 5 | 1 | 1 | 2 | 3 | 1 | 4 | 2 | 5 | 4 | 37 | 156 |
| 2011 | 0 | 4 | 1 | 3 | 0 | 2 | 1 | 3 | 3 | 3 | 2 | 3 | 25 | 148 |
| 2012 | 2 | 1 | 3 | 4 | 6 | 2 | 2 | 1 | 2 | 5 | 2 | 2 | 32 | 170 |
| 2013 | 2 | 0 | 2 | 0 | 2 | 2 | 4 | 2 | 1 | 1 | 4 | 3 | 23 | 148 |
| 2014 | 0 | 1 | 4 | 4 | 4 | 6 | 6 | 0 | 3 | 1 | 1 | 5 | 35 | 179 |
| 2015 | 3 | 0 | 1 | 4 | 4 | 1 | 3 | 1 | 6 | 1 | 5 | 1 | 30 | 165 |
| 2016 | 2 | 1 | 6 | 4 | 6 | 1 | 3 | 2 | 6 | 3 | 6 | 1 | 41 | 111 |
| 2017 | 4 | 1 | 2 | 2 | 1 | 4 | 2 | 4 | 2 | 3 | 1 | 1 | 27 | 138 |
| 2018 | 0 | 2 | 0 | 3 | 4 | 4 | 1 | 0 | 0 | 2 | 0 | 0 | 16 | 175 |
| Total | 76 | 84 | 84 | 101 | 110 | 80 | 88 | 81 | 92 | 105 | 93 | 89 | 1083 | |

2018 [58] (Table 1). The field studies conducted once or twice every year in Moange le Bosquet from 2007 until 2019 included questionnaires about family composition.

In order to assess the potential link between environmental conditions and the reproductive pattern, we used data from the ERA5 report provided by the European Centre for Medium-Range Weather Forecasts (ECMWF, https://www.ecmwf.int). ERA5 provides hourly estimates of a large number of climate variables, covering the Earth on a 30km grid. ERA5 compiles vast amounts of historical observations into global estimates using advanced modelling and data assimilation systems [64]. Data available from the closest weather station to Moange le Bosquet are from the village of Messok, 35 km to the east. The wet seasons occur in March-May and September-November and the dry seasons in December-February and June-August. The meteorological data include precipitation and temperature from 1978 to 2018.

## Methods of data analysis

**Metric of reproductive seasonality.** In order to assess reproductive seasonality, we chose to use the classic metric of birth amplitude [e.g. 12, 35] defined as the *percentage deviation in the number of births from the annual monthly mean* and hereafter written as *Dev.Birth*. This metric has the advantage of de-trending potential year-to-year fluctuations in numbers of births [18, 70]. It was calculated as the difference between the number of births in specific months and years (e.g. number of births in January 1992) and the mean number of births per month over the same year (e.g. average monthly births in 1992), divided by the monthly mean for the year. *Dev.Birth* is month and year specific and is obtained with the following equation:

$$\text{Dev.Birth}_{ij} = \frac{N_{ij} - M_j}{M_j},$$

where *N* is the number of births in a given month (i) of a given year (j) and M the mean number of births per month in a given year (j).

Due to the small sample size, the data show wide variations in the number of observed births across years and months (mean = 2.65, SD = 1.78, ranging from 0 to 8). On the one hand, we wanted to keep the variance between years to account for temporal fluctuations in births between years, but on the other hand, the small sample size implies wide fluctuations in the observed number of births for a given month of a given year. To capture potential changes in births patterns across years while decreasing the noise resulting from sampling variance, we median-smoothed the number of births in a given month across three successive years (with the 'smooth' function in the 'stats' package in 'R' (R Core Team, 2020), using the '3' methods and a 'copy' end rule). We then calculated the percentage of monthly deviation in relation to the annual mean.

Due to the loss of birth records between 1983 and 1987, some of the data were missing. For descriptive statistics and correlation tests, we merged the data for 1982 with the 1988 data (as if there was no gap within those years) but not for trends analyses across years (where only the years from 1988 to 2018 were kept).

As indicated above, the number of women of child-bearing age is expected to be constant through years. However, because changes in the number of births across years can be due to differences in the number of women present in the village, we further investigated this potential bias. Our results confirmed that the data for the number of births are valid to study reproductive patterns in this population (S1 Text, S1 Fig).

**Seasonal fluctuation in births.** To test for seasonal fluctuations in birth patterns we performed sinusoidal regressions so that:

$$X_t = \mu + a\,\cos(ft) + b(ft) + e_t$$

Where $X_t$ is the percentage deviation in number of births from the annual monthly mean (*Dev. Birth*), *t* is the month of the year, and *f* is the frequency of periodic variation. Sinusoidal models such as these have been previously used to characterize the seasonality of various events [e.g. 4, 71, 72]. More specifically, we tested two wavelengths: 2π/12 and 2π/6 corresponding respectively to a unimodal or bimodal distribution of *Dev.Birth* per year.

**Reproductive seasonality and climatic variables.** Data for precipitation and temperature for each month of each year are, respectively, the monthly mean daily precipitation (in mm) and the monthly mean temperature (in C°) (S1 and S2 Tables).

We used linear regression to investigate whether four explanatory variables–precipitation (daily mean for each month in mm, S1 Table) and temperature (monthly mean in C°, S2 Table) at birth and conception (taken as 9 months before birth)–explain reproductive seasonality (Dev.Birth) within the year. More specifically, we tested all potential models combining the four explanatory variables and their interactions together with the mean temperature and precipitation per years using the 'dredge' function of the 'MuMIn' package in 'R' and we ranked them as a function of the AICc [73]. Among the best models (ΔAICc<2), we chose the most parsimonious in terms of degree of freedom.

**Change across years in the timing of seasonal birth peaks.** As an initial investigation of potential change in reproductive seasonality over time, analyses were performed for the years before 2003, which represent half of our time series, and for the years 2003 and after. As some differences were found between these two samples (Fig 2) an analysis in-depth was performed. To assess whether the birth pattern might have changed over the years, we analysed the potential changes in the months for which *Dev.Birth* was at its minimum and maximum each year from 1988 to 2018. We chose this method because we suspected birth patterns to be bimodal with possibly different changes in the timing of the two peaks across years, which would be difficult to detect using harmonic regression. We indexed the months by values from 1 to 12. For each year, we then looked for the month(s) where *Dev.Birth* was at its minimum between the months of November and February (hereafter written as 'Minimum 1') and between May and August (hereafter written as 'Minimum 2'). These were chosen based on the timing of the troughs in the bimodal distribution revealed by previous analysis (see Results below). When there were several equal minimum values for these periods, the mean indexes for the months were used. Finally, we looked for the maximum values in between these two consecutive minimum values, thus locating the two corresponding birth peaks (hereafter written as 'Maximum1' and 'Maximum 2') in relation to the birth minimums. To assess changes in the timing of these minimums and maximums, we performed linear regressions of the timing of these yearly troughs and peaks for the years from 1988 to 2018.

## Results

### Birth seasonality

The *percentage deviation in the number of births in the annual monthly mean* (*Dev.Birth*) before and after year 2003 (the mid-point of the following period) is shown in Fig 2. Although there is more variation across years in years 1980–1982 and 1988–2003 than in years 2003–2018, Fig 2A and 2B both exhibit a seasonal distribution of births across the year with two modes. For years 1980–1982, 1988–2002 the mean maximum deviations are +20% respectively for the first (in April) and second (in August) peaks and -20% (in January) and -38% (in June)

A

**Years 1980−1982, 1988−2002**

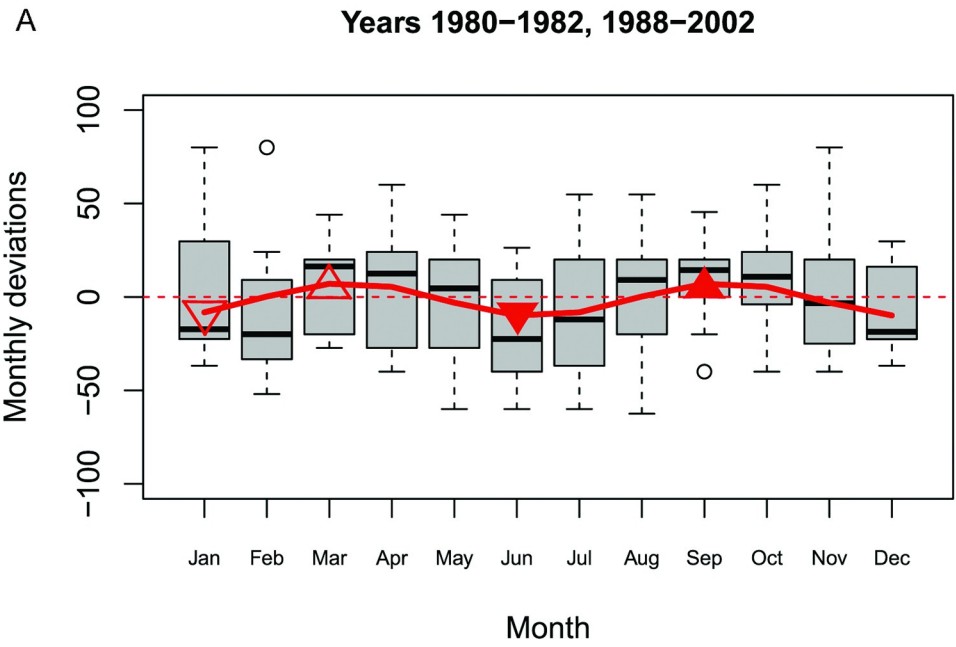

B

**Years 2003−2018**

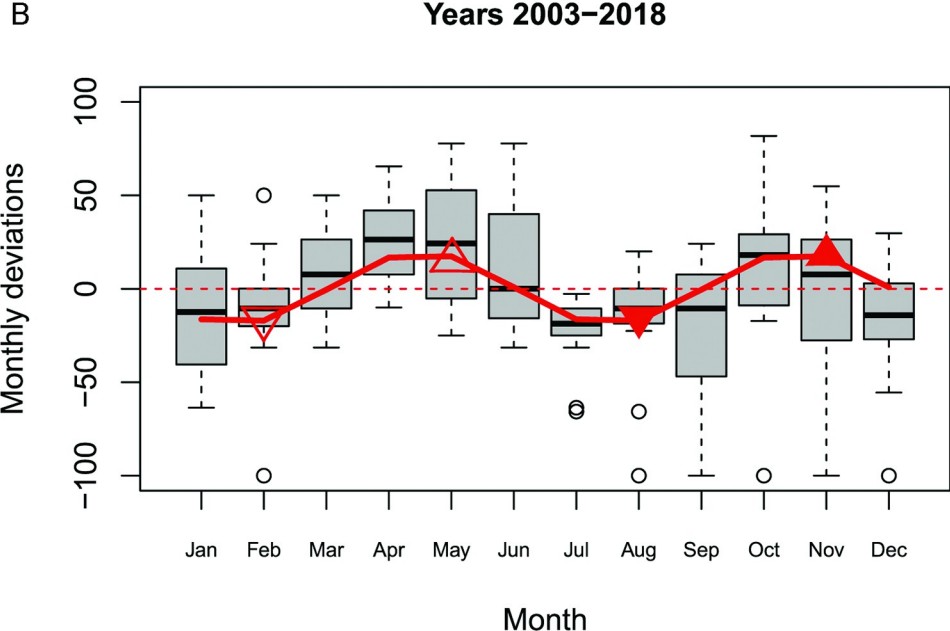

**Fig 2. Births monthly deviation.** Boxplots showing Dev.Birth median-smoothed across three successive months for the years 1980–1982 and 1989–2003 (A) and 2003–2018 (B) and the corresponding fitted sinusoidal regression shown by the red line. The average peak and trough months are marked and determined as: 'minimum 1' (lowest number of births between November and February,▽), 'maximum 1' (highest number of births between 'minimum 1' and 'minimum 2', △), 'minimum 2' (lowest number of births between May and August, ▼) and 'maximum 2' (highest number of births between 'minimum 2' and 'minimum 1',▲).

respectively for the first and second minimums. For years 2003–2008, the first maximum (in April) is +38% and the second maximum (in October) is +29% while the first minimum (in January) is -18% and the second minimum (in August) is -48%. These deviations are statistically significant as tested by harmonic regression. This is further confirmed by sinusoidal regression favouring a wavelength of $2\pi/6$ with p = 1.834e-05 ($a$ = -4.18 and $b$ = -10.75) for the years before 2003 and for 2003 and after, with p = 1.462e-05 ($a$ = O.65 and $b$ = -19.47) for the years after 2003. It must be stressed that the magnitudes of these deviations is not small. For instance, at a whole other scale–that of country or ecological zone–maximum deviation observed by Dorélien [12] range from +12% to +49%.

The month of birth accounts for 8.4% of the variance in *Dev.Birth* (respectively 4.6% before 2003 and 11% in 2003 and after). S2 Fig shows similar results for the monthly number of births over the whole period, and S1 Fig shows similar results for monthly fertility rates for the years for which we have a good estimate of women of reproductive age, showing that potential inter-annual fluctuations in the number of women do not bias our results (S1 Text).

The two bimodal distributions show an initial negative deviation in November-February before 2003 and in December-January after 2003. The first wave of positive values is in March-April for the years before 2003 and in April-May for the years after 2003. The second wave of negative values is in June-July for the years before 2003 and in July-August after 2003, followed by a second wave of positive values in August-October for the years before 2003 and in October-November for the years after 2003. Assuming a 9-month pregnancy, these results suggest that the corresponding peaks in conceptions would be in June-July and November-December for the years before 2003 and in July-August and January-February for the years after 2003.

## Reproductive seasonality and climatic variables

We ran 452 models testing the effects on Dev.Birth of combinations of the four explanatory variables (precipitation and temperature at conception and at birth) and their interactions together with the mean annual temperature and precipitation. Of these models, 12 were ranked as best ($\Delta$AICc<2). Among these, two alternative models are equally parsimonious, incorporating precipitation and temperature either at birth or at conception (presented in Table 2). Using monthly deviations from annual precipitation and temperature means as explanatory variables gives similar results (not shown). Our results demonstrate a significant positive association of precipitation and temperature with birth and a negative association of precipitation and temperature with conception (Table 2).

**Table 2. Linear regression between *Dev.Birth* and climatic variables at birth and conception.**

| Predictors | Model 1 | | | Model 2 | | |
|---|---|---|---|---|---|---|
| | Estimates | CI | p | Estimates | CI | p |
| (Intercept) | -175.08 | -326.12–-24.05 | **0.023** | 185.23 | 34.62–335.85 | **0.016** |
| prec.birth | 4.26 | 1.56–6.97 | **0.002** | | | |
| temp.birth | 6.71 | 0.51–12.92 | **0.034** | | | |
| Prec.conc | - | - | - | -3.98 | -6.65 –-1.31 | **0.004** |
| Temp.conc | - | - | - | -7.19 | -13.38 –-0.99 | **0.023** |
| Observations | | | 408 | | | 408 |
| R2 / R2 adjusted | 0.025 / 0.021 | | | 0.024 / 0.020 | | |

prec.: precipitation, temp.: temperature, conc: conception.

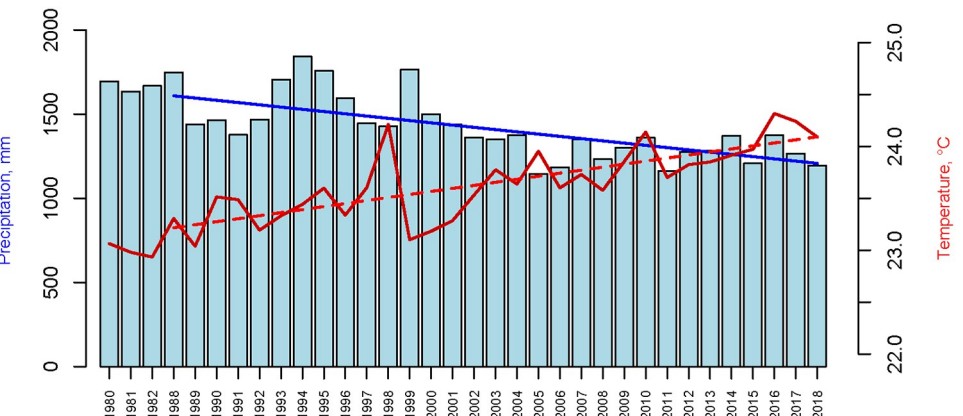

**Fig 3. Total amount of yearly rainfall (mm) and daily mean temperature (ºC) for years 1979–2018.** The regression lines highlight the decrease in rainfall (p = 2.13e-6) and the increase in temperature (p = 1.16e-8). Data from ERA5 [64].

### Change across years in the timing of seasonal birth peaks

Detectable changes in rainfall and temperature have been recorded over the last 30 years in the region of Le Bosquet. Weather data from 1979 to 2018 show an overall decrease in precipitation of about 30% and an increase in temperature of about 1.15ºC (Fig 3). These changes have affected the magnitude of these weather variables but have not modified their pattern of occurrence over the year, which is observed to be the same before and after year 2003 (Fig 1).

As explained in the Fig 2 legend, 'minimum 1' is the lowest number of births between November and February, while 'minimum 2' is the lowest number of births between May and August. Then there is 'maximum 1' which is the highest number of births between 'minimum 1' and 'minimum 2' and also 'maximum 2' as the highest number of births between 'minimum 2' and 'minimum 1'. Linear regression on the timing of 'minimum 1' and 'maximum 1' show no changes in their timing over the years (p>0.1, results not shown) (S3 Fig). However, in the last 30 years, the timing of 'minimum 2' and 'maximum 2' has shifted significantly to later in the year (p<0.01, Fig 4), roughly from late May to mid-July and from late August to late October respectively. Thus, the distance between the two peaks tends to increase, and subsequently, the second peak and the subsequent first one of the next year is decreasing. Applying a similar method to precipitation and temperature data confirmed that, conversely, there were no significant changes in the timing of precipitation and temperature over the years. But our results show a decline and an increase, respectively, in the magnitudes of precipitation and temperature from 1979 to 2018 (but with no change in their timing, see Fig 3 and S1 Table). These changes occur in parallel with a change in the timing (but not in the magnitude) of births in the second part of the year.

### Discussion

This study had several results: 1) the presence of reproductive seasonality among the Baka; 2) the relationship between reproductive seasonality with meteorological variables, and 3) local evidence of climate change in southeastern Cameroon and its relationship with the Baka reproductive pattern. We discuss these findings with regard to the socio-economic activities of the Baka.

Our analyses demonstrate significant two-peak seasonality in births among the Baka, a not uncommon pattern among several societies [e.g. 23, 30, 31]. The two peaks occur at an interval

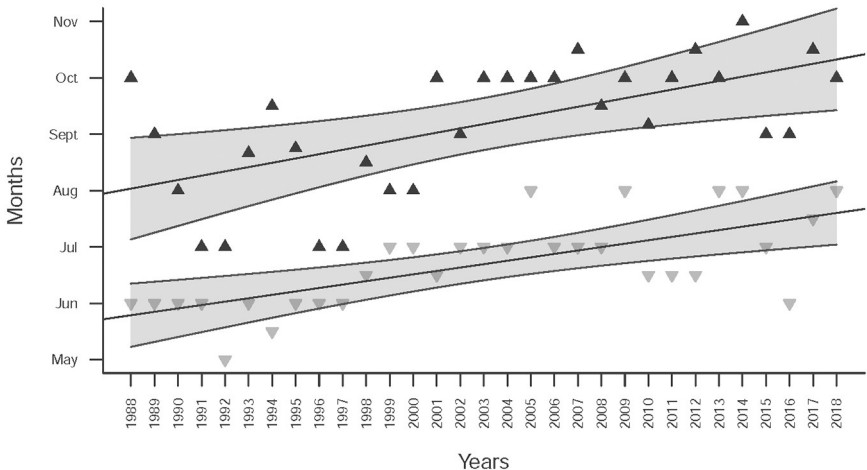

**Fig 4. Extremes of births monthly deviation.** Months when *Dev.Birth* is at a minimum between May and August (i.e., 'minimum 2', ▼) and at a maximum between the two minimums (i.e., 'maximum 2', ▲). The linear regression lines have coefficients of 0.06048 (p-value = 1.49e-5) for 'minimum 2' and 0.0764 (p-value = 0.000321) for 'maximum 2'. The grey areas show the 99%CI confidence intervals of mean predictions.

of about 6 months (Fig 2), in contrast with other studies where the two peaks are separated by 4–5 and 8–7 months respectively [23, 30, 31]. The modal 2.5-year IBI for the Baka is equivalent to five 6-month cycles. This means that successive sibling births will occur alternately in one peak or the other. The shift over time in the second peak would probably accompanies a change in the IBI. IBI data covers a shorter period than the records of birth, thus possible changes in the IBI over time are more difficult to assess. However, Ramirez Rozzi [58] has observed that the average IBI for mothers 15–25 years of age is 2.36 years in 2010 whereas it is 2.56 years in 2014. This change is not significant (P = 0.384) [58].

The two-peak pattern correlates well with rainfall fluctuation over the year. Multivariate analysis demonstrates that precipitation at conception or at birth potentially have effects, respectively negative and positive, on the monthly number of births. Note that because we do not have any information on the number of conceptions that do not result in births, we cannot perform two sets of analyses for, respectively, the number of conceptions and births per month. We were not able to distinguish whether these are independent effects or whether they arise because of the co-linearity with the climatic variables. In both cases, however, there is a combined mechanism at work between temperature and rainfall, as found by others [3]. In this sense, temperature might not correlate directly with conceptions and births (via biological or behavioural mechanisms) but it might have a role in controlling other variables that do affect the reproductive pattern. Together, the climatic variables accounted for 30% of the variance in *Dev.Birth* explained by the month of birth.

Health, nutrition and social issues related to specific seasons could provide us with potential explanations for the occurrence of these peaks in conception in the dry seasons (Fig 1). Conception outcomes are better in dry periods because the incidence of infectious diseases is lower than during rainy periods in tropical regions [22, 29, 31]. Berry et al. [74] have shown that in regions where malaria is endemic, the prevalence of malaria is greater in women who become pregnant during the rainy season than during the dry season. Specific data on the incidence of malaria, or any other infectious disease, do not exist for the Baka. However, Cameroon has a relatively high prevalence of malaria [75] and the Baka are affected by this disease, as suggested by the number of plants and remedies they know for treating it [76], so we cannot omit this possible determinant.

A good nutritional status has been pointed out as a key factor for higher conception rates [12, 21, 22, 31]. Although our study has not collected actual data on energy intake, our knowledge on time allocation, dietary diversity and resource availability among the Baka does allow speculation as to the energy available over the year [66, 77]. While there is no record of severe starvation among the Baka [58], dry seasons are likely to ensure a better nutritional status than the rainy seasons, when the Baka diet is less varied with less frequent consumption of starchy staples, meat, fruits and vegetables [66]. Specifically during the dry seasons, the overall availability of wild food is greater [i.e. 58, 59]. For instance, wild yams (*Dioscorea sp*.), one of the main wild sources of carbohydrates and calories for the Baka [59], are larger and consumed more frequently during dry seasons, while in rainy seasons, yam tubers are small and only consumed in cases of necessity [59]. The yam preferred by the Baka, *D. praehensilis*, is most available in the wild in the month of December [77]. More specifically regarding the intake provided by the food during the dry seasons, the less intense dry season, in which we found the August conception peak, is characterized by a higher intake of bush mango [61], which can provide up to 50% of the calorie intake [68] thanks to the high proportion of lipids in the fruit [78]. During the more intense dry seasons, which matches the first conception peak in January wild meat and fish are more frequently consumed than in the other seasons [65]. The inclusion of fish, wild meat and larger amounts of starchy yam might improve the nutritional status of the population during this period. Conception peaks in the Baka thus seem to be concentrated in periods of better nutrition. These insights are consistent with two previous studies which found that conception patterns are related to better nutritional status among hunter-gatherers [36, 37] and additional studies on agricultural societies [12, 13, 30, 31].

Other socio-cultural factors could play a role in reproductive seasonality in the Baka, such as mobility patterns and subsistence and social activities, which are directly relevant to physical and emotional proximity among the Baka and may thus affect the frequency of intercourse in the population [79]. The fact that conception rates are low during rainy seasons is not surprising if we consider the mobility pattern of the Baka. In these periods, men are more frequently involved in hunting expeditions (especially during the more intense rainy season, from September to November). While they are away, the women are symbolically involved in the success of the men's hunting and have to respect socio-cultural practices and norms in order to avoid hunting failure, including avoidance of extramarital relations [42]. During the longer rainy season, the Baka are also more engaged in harvesting cacao beans for their Bantu neighbours [80]. During this period, the Bantu hire the Baka, mostly men, for several days to several weeks, taking them to their villages. The women and children sometimes join their men in the Bantu villages but sometimes stay home, which therefore also affects the couple's interactions. In contrast, during the dry seasons, families are more likely to be together. This proximity could be even greater during the more intense dry season, as Baka households tend to live in their forest camps in small numbers. The small size of these camps (30–50 individuals), made up of close relatives, might give the couples more intimacy. As privacy could favour more frequent sexual intercourse [81], more conceptions would be likely during this period. In August, however, the Baka tend to be more present in the village, mostly to dig new farm plots and carry out agricultural tasks [56]. Gathering expeditions to the forest and forest camps take place, but much less frequently than in the other dry season. Therefore, in the Baka lifestyle, mobility patterns related to subsistence and social activities could play a role in reproductive seasonality.

Other studies have shown changes in the magnitude [69, 79] or the shape of the birth pattern over time [11]. Our study also shows such a change: a decrease/increase in precipitation/temperature within the last 30 years parallel to a shift in the timing of the second peak of births in the year. Although we do not demonstrate causality between these two effects, we can

hypothesize that a change in food availability due to the decline in precipitation may affect the dynamics of subsistence activities or of the energy balance, which in turn has an impact on the timing of births. For instance, the decline in precipitation through the entire year may have a greater effect on food availability during the more intense dry season—producing a clear impact on births at the second peak occurring 9 months later–and a lesser effect in the less intense dry season (June and July) producing a negligible effect on the first peak. Whatever the reason, these results appear to suggest that the impact of climate change has already started to affect hunter-gatherer life patterns.

In conclusion, this study has shown that there is a clearly seasonal pattern of reproduction among the Baka with two main peaks of births each year, that conceptions mainly occur during the dry seasons and, finally, that a shift in the peak of births seems to relate to climatic changes over the years.

## Supporting information

**S1 Text. Number of births is a valid method to assess seasonality in births.**
(PDF)

**S1 Table. Precipitation and average daily precipitation by month and year (mm).**
(PDF)

**S2 Table. Temperature and average daily temperature by month and year (˚C).**
(PDF)

**S3 Table. Fertility rates per month and year, using monthly numbers of births and yearly numbers of women.**
(PDF)

**S1 Fig. Monthly deviations from annual averaged fertility rate (A) and from the annual averaged number of births (B) for the 2007–2018 period.** For each set of values (A and B), two types of graph were produced, boxplots (left) to show the raw data distribution and locally weighted polynomial plots (right) generated with the 'locally.weighted.polynomial' of the SiZer library in 'R', to show smoothed data.
(PDF)

**S2 Fig. Number of births per year (upper panel) and boxplot of the number of births across months (lower panel).** The red line stands for the sinusoidal regression of wavelength of $2\pi/6$ with p = 0.02272 (a = -0.04412 and b = -0.33962). The number of births per year (showing that it varies from 16 to 55 recorded births) and the raw data variations in the number of births from month to month analyzed by harmonic regression confirm that birth seasonality is significantly bimodal.
(PDF)

**S3 Fig. Extremes of birth month deviations for the first half of the year.** Months when *Dev. Birth* is at a minimum between November of the previous year and February (i.e., 'minimum 1', down-pointing triangle) and at a maximum between the two minimums (i.e., 'maximum 1', up-pointing triangle). The linear regression lines have non-significant coefficients (p-value>0.1).
(PDF)

## Acknowledgments

Thanks to X. Garde, B. Bordage and to all the staff of IRD Yaounde for their logistical support; to the nuns at the Le Bosquet mission for their kind hospitality and especially to P. Kalo, J. B. Etoa and other Baka collaborators for their help, assistance and friendship. We also want to thank A. Epelboin for his useful comments on the previous manuscript; Elisa Bergas Massó, who provided us with high quality meteorological data without which the study could not be carried out, and Mikel Larrañaga Arcay. This study was supported by the CNRS and IRD, and by grants from the Wenner–Gren Foundation (7810) (FRR), the National Geographic (8863–10) (FRR) and the Agence Nationale de la Recherche under the Blanc SVSE7-2011 GrowinAP programme (FRR).

## Author Contributions

**Conceptualization:** Samuel Pavard, Fernando V. Ramirez Rozzi.

**Data curation:** Fernando V. Ramirez Rozzi.

**Investigation:** Laura Piqué-Fandiño, Fernando V. Ramirez Rozzi.

**Methodology:** Sandrine Gallois, Samuel Pavard.

**Supervision:** Samuel Pavard, Fernando V. Ramirez Rozzi.

**Validation:** Samuel Pavard, Fernando V. Ramirez Rozzi.

**Writing – original draft:** Laura Piqué-Fandiño, Sandrine Gallois, Samuel Pavard, Fernando V. Ramirez Rozzi.

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
