## [Decision Letter · Decision Letter 0]

25 Aug 2021

PONE-D-21-22413

Seasonality of birth in the Baka Pygmies, environmental factors and climatic changes

PLOS ONE

Dear Dr. Ramirez Rozzi,

Thank you for submitting your manuscript to PLOS ONE. After careful consideration, we feel that it has merit but does not fully meet PLOS ONE’s publication criteria as it currently stands. Therefore, we invite you to submit a revised version of the manuscript that addresses the points raised during the review process.

ACADEMIC EDITOR: Thank you for submitting your work to Plos One. We now have two complete reviews, and both reviewers find that the manuscript has the potential to meet Plos One criteria for publication. However, both reviewers raise questions about the statistical analysis and the availability of data, as well as code for reproducing the analysis. These questions must be addressed prior to acceptance for publication. In particular, the authors must provide access to the data and code used to run the analysis, preferably as electronic files, to improve the ability of reviewers and readers to evaluate the data and use the remarkable data presented by this study. I realize that tables are presented in the word document SI. However a more analysis friendly format would be helpful. Further, the statistical analysis requires some more justification and attention to the interpretation of the results in terms of effect size. I would also note that mutual information is often a more appropriate metric for assessing the association between a climate and a population variable in a time-series when such series are short and noisy. See, for example, Cazelles B (2004) Symbolic dynamics for identifying similarity between rhythms of ecological time series. Ecol Lett 7:755–763. R1 and R2 provide additional comments that may improve the quality of the manuscript. For your reference, files are attached with R1 and R2 comments.

We look forward to receiving your revised manuscript.

Kind regards,

Jacob Freeman

Academic Editor

PLOS ONE

Journal Requirements:

2. Please complete the following questions designed to promote ethical global research practices. The information requested in this section should also be included in the Methods section of your manuscript. 

Did you obtain permissions from a local agency (e.g. government office, ethics committee or similar) to conduct this study? 

Provide details as to who granted local permissions and/or consent. Refer to any individuals involved by their role or title but do not list their name(s).

Did you obtain written informed consent from a representative of the local community or region before the research took place?

Did the individuals who provided consent for this research also agree to the sharing of the data obtained, as per your Data Availability Statement? 

How were the aims of the research investigation, its methodology, and its anticipated outcome(s) discussed with, and agreed upon by the people/s being studied or representatives of the local community?  How did you establish who speaks for the community?

What aspects of the research process, including the right to exercise control over how the information and/or materials are collected, studied, and/or published, did the local community being studied have control over? 

How were the ethical values of the local community acknowledged, discussed, and/or incorporated into the research design?

Have the findings of the research been presented or made available in an understandable format to stakeholders in the community where the study was conducted (e.g. via a presentation, summary report, copies of publications, etc.)?

3. We note that you have referenced (ie. Bewick et al. [5]) which has currently not yet been accepted for publication. Please remove this from your References and amend this to state in the body of your manuscript: (ie “Bewick et al. [Unpublished]”) as detailed online in our guide for authors

Reviewers' comments:

Reviewer's Responses to Questions

**Comments to the Author**

1. Is the manuscript technically sound, and do the data support the conclusions?

Reviewer #1: Yes

Reviewer #2: Yes

2. Has the statistical analysis been performed appropriately and rigorously? 

Reviewer #1: Yes

Reviewer #2: I Don't Know

3. Have the authors made all data underlying the findings in their manuscript fully available?

Reviewer #1: No

Reviewer #2: Yes

4. Is the manuscript presented in an intelligible fashion and written in standard English?

Reviewer #1: Yes

Reviewer #2: Yes

5. Review Comments to the Author

Reviewer #1: Reviewed by M. Clauss:

This manuscript presents data and analyses of reproductive seasonality in a group of subsistence hunter-gatherer-farmers-labourers, the Baka Pygmies. The data is sound (but not available easily for others - if one would want to repeat the analyses, one would miss the climate data at the monthly resolution - these need to be supplied).

My main concern is the narrative of the manuscript. I made ample comments on this in the attached WORD file of the manuscript and ask the authors to go through those comments. I sum up some points here.

1. The justification of the work: this is done by (i) invoking relevance for the 'threat' status of this population (due to its life style), but this thread is not taken up in the discussion, and the relevance of the knowledge for a protection or for the benefit of the population is not even mentioned any more. (ii) by claiming important links of birth seasonality with other health parameters (like longevity, neonate survival etc.) - which are then never again mentioned. This sums up like a typical narrative that claims to be relevant for certain reasons but then fails to address these issues in the discussion. For me, this is one reason why people do not trust scientists - because many scientists use cool-sounding justifications for their work but then do not keep the promise these justifications made. It would be more honest, and satisfying, to give other reasons, and to mention these 'justifications' as outlook in the discussion (e.g., saying further work could test an effect of the seasonality on health/survival).

2. The structure of the explanations: In my view, the results are very similar to reference 30 Philibert et al. 2013, but this paper is only very sporadically referred to. In the discussion (and partly in the results), the authors give reasons why one would expect a seasonal reproductive pattern in this population - but this feels like an afterthought. It would be better to outline possible reasons for reproductive seasonality in the Intro and derive predictions from that. In particular, the discussion indicates that there are 3 different possible reasons for the observed seasonality, which the data alone cannot distinguish. I would find the manuscript much more satisfying, and honest, if these predictions were outlined at the beginning, also mentioning that one cannot distinguish between them.

3. The description of the lifestyle. The Baka are called "hunter-gatherers" especially in the Abstract and Intro, and only later it is stated that they also practice some kind of subsistence agriculture, and hire out as labourers to other groups. This comes as a disappointment, after the strong claims about "hunter-gatherers". I think the complete lifestyle picture should be mentioned in the abstract, and the Intro / Methods.

4. Structuring causes for reproductive seasonality. Coming from mammals, the main cause is nutritional status and health status (this are factors basically common to all animals). In humans, there are additionally behavioural and socio-economic factors. It would be good to clearly structure these effects in the Intro, e.g. there are

(i) constraints on conception or pregnancy maintenance due to nutritional status, or infectious diseases (Malaria), or effects of climate on sperm quality

(ii) constraints on the frequency of mating / intercourse, due to behavioural effect of climate (discomfort), of workload (tiredness in certain seasons), of working patterns (e.g. working or hunting away from mates), cultural restrictions, and deliberate birth timing.

(iii) evolutionary ingrained triggers (e.g. phototriggers) that probably play a lesser role in humans in general, but also have little effect in the tropics.

In the population here, many causes for reduced conception/mating seem to co-incide in the wet seasons.

5. Mentioning the magnitude of effect. Statistical significance is a necessary condition for the relevance of a finding, but more important is the magnitude of effect. Depending on how one calculates it, we get about 10% of all births in this population being due to seasonality, as opposed to about 90% being due to non-seasonal reproduction. This is never clearly stated in the manuscript, and is not part of the discussion about the reasons. But in my interpretation, stating this clearly would emphasize that while all the mentioned effects may have some effect, they are not overriding rulers of the Baka's reproduction. If you disagree about this conclusion, you should mention it and explain why it is wrong. Not mentioning the magnitude is just strange.

6. There are several specific instances where the text seems to contradict itself in short sequence, e.g. when saying that in the dry season, the diet is more diverse, but a few sentences later is is said that it is made up of 50% of one single kind of fruit. If it is dominated by a single fruit and YET more diverse, you shoud explain this so the reader does not feel confused. Another example is to say that the lack of such studies is "surprising" and then immediately explain why this data is hard to come by. If data are hard to come by, lack of work on it is not really a surprise .... Other instances indicated in the word file.

7. I would expect to have a clearer overview over the literature in terms of 'double peaks', e.g. Philibert et al. 2013 (I did not check other human literature).

8. The finding about an effect of global warming on the reproductive seasonality says that there is an effect on the second birht peak in the year (which occurs at later dates) but not on the first. This clearly means that the distance between these two peaks is increasing (or, in other words, the distance between the second peak and the subsequent first one of the next year is decreasing). This is not mentioned, and not analyzed. In particular, this should affect discussions about the "contstant" interbirth interval and the sentence that "siblings are likely to be borne at subsequent peaks". For example, one could correlate the dates of the peaks to test whether their distance increases, or correlate that distance with the years. Maybe in years where the second peak is particularly late, the next first peak of the following year is also delayed?

9. Very generally, the manuscript makes it clear that you do distinguish between birth and conception. Nevertheless, the choice of words often is as if you claimed that there is a direct effect of something, like temperature, on birthing. It is clear this is not how you mean it, yet the words sound like it. I recommend to mend this. One good way would be to replace the term "birth seasonality" by "reproductive seasonality" in many locations.

Please see the attached word file for details. My apologies for some strong criticism, but I hope this gives you an idea how your work in its present state might be regarded.

sincerely marcus clauss

Please refer

Reviewer #2: See attached PDF.

This form requires more characters for some reason, so here are some:

"Sed ut perspiciatis unde omnis iste natus error sit voluptatem accusantium doloremque laudantium, totam rem aperiam, eaque ipsa quae ab illo inventore veritatis et quasi architecto beatae vitae dicta sunt explicabo. Nemo enim ipsam voluptatem quia voluptas sit aspernatur aut odit aut fugit, sed quia consequuntur magni dolores eos qui ratione voluptatem sequi nesciunt. Neque porro quisquam est, qui dolorem ipsum quia dolor sit amet, consectetur, adipisci velit, sed quia non numquam eius modi tempora incidunt ut labore et dolore magnam aliquam quaerat voluptatem. Ut enim ad minima veniam, quis nostrum exercitationem ullam corporis suscipit laboriosam, nisi ut aliquid ex ea commodi consequatur? Quis autem vel eum iure reprehenderit qui in ea voluptate velit esse quam nihil molestiae consequatur, vel illum qui dolorem eum fugiat quo voluptas nulla pariatur?"

6. PLOS authors have the option to publish the peer review history of their article (what does this mean?). If published, this will include your full peer review and any attached files.

Reviewer #1: **Yes: **Marcus Clauss

Reviewer #2: **Yes: **Edward Hagen

---

## [Author Response · Author response to Decision Letter 0]

16 Nov 2021

Reviewr 1

Reviewer #1: Reviewed by M. Clauss:

This manuscript presents data and analyses of reproductive seasonality in a group of subsistence hunter-gatherer-farmers-labourers, the Baka Pygmies. The data is sound (but not available easily for others - if one would want to repeat the analyses, one would miss the climate data at the monthly resolution - these need to be supplied).

The data is fully available. The data about birth is presented in Table 1 and data about climate, as it is indicate in our ms, comes from the ERA5 report provided by the European Centre for Medium-Range Weather Forecasts (ECMWF, https://www.ecmwf.int). The clear evidence that the data is fully available is that the Reviewer #2 re-run all our analysis and his remarks come from this re-run (see Reviwer# 2 comments and answers).

My main concern is the narrative of the manuscript. I made ample comments on this in the attached WORD file of the manuscript and ask the authors to go through those comments. I sum up some points here.

We thank the reviewer for his recommendations. The changes from his comments are in the marked-up copy of our manuscript.

1. The justification of the work: this is done by (i) invoking relevance for the 'threat' status of this population (due to its life style), but this thread is not taken up in the discussion, and the relevance of the knowledge for a protection or for the benefit of the population is not even mentioned any more. (ii) by claiming important links of birth seasonality with other health parameters (like longevity, neonate survival etc.) - which are then never again mentioned. This sums up like a typical narrative that claims to be relevant for certain reasons but then fails to address these issues in the discussion. For me, this is one reason why people do not trust scientists - because many scientists use cool-sounding justifications for their work but then do not keep the promise these justifications made. It would be more honest, and satisfying, to give other reasons, and to mention these 'justifications' as outlook in the discussion (e.g., saying further work could test an effect of the seasonality on health/survival).

The reviewer’s comment on this point on the ms is very interesting. The criticism he makes, it is true, can be valid but it is also based on erroneous interpretations of what we have written. Because it says “and here you become implicit – this is needed to help protect the group” and at no time are we talking about helping or protecting the group. Although the sentence may have been awkward and has been removed from this version, the interpretation given by the reviewer goes beyond what we are talking about. Knowledge of the RS in this population is not oriented to their protection or benefit. It is a descriptive work to understand a situation that is likely to change (lines 139-169). We do not see how to apply a value judgment that would lead to the protection of a RS. Even we can start from the assumption that the RS we observe is the best because it has been shaped by years of interaction with the environment. Moreover, to approach the protection or benefit of a particular situation from our perspective, without knowing or taking into account the will of the Baka would be a bit haughty.

The relationship that reproductive seasonality presents with other life history variables is not presented in this ms, it is not the objective of the study. Many authors in previous work have suggested the relationships between life history variables and reproductive seasonality. This paragraph is here to highlight the importance of this type of analysis in a given population. It is true that neither the birth seasonality nor the other health parameters are known in the Baka and thus this work represents the first step to know how these relationships between RS and other health parameters occur in the Baka.

Moreover, that climate change is affecting reproductive seasonality is not seen as a problem, it is the observation of a situation that deserves to be monitored.

2. The structure of the explanations: In my view, the results are very similar to reference 30 Philibert et al. 2013, but this paper is only very sporadically referred to.

Philipert et al. focused their work on rural societies in Mali. Since the environmental conditions are so differently influencing the everyday life in rural societies and hunter-gatherer societies, we restraint comparison of the Baka with other hunter-gatherer societies.

In the discussion (and partly in the results), the authors give reasons why one would expect a seasonal reproductive pattern in this population - but this feels like an afterthought. It would be better to outline possible reasons for reproductive seasonality in the Intro and derive predictions from that. In particular, the discussion indicates that there are 3 different possible reasons for the observed seasonality, which the data alone cannot distinguish. I would find the manuscript much more satisfying, and honest, if these predictions were outlined at the beginning, also mentioning that one cannot distinguish between them.

Thank you for this comments. We present now in the Introduction, based on previous works on RS, the possible reasons which can lead to suppose that RS exists in the Baka. It can be read from line 140 to line 168, as followed “Since previous studies of the Ache in the rainforests of eastern Paraguay have demonstrated birth seasonality [29], we expect the Baka to show reproductive seasonality as well. Studies on reproductive seasonality have found birth peak amplitudes ranging from 5 to 65% [12, 23, 27, 29]. Recent studies on growth and life history variables in the Baka Pygmies in Cameroon [58, 67] have shown a modal interbirth interval (IBI) of 2.5 years. Since the IBI is 2.5 years, successive siblings are born at two different times in the year. If a peak marks a seasonal birth pattern in the Baka, we would expect a second peak six months later and thus a two-peak pattern of birth seasonality. Two-peak reproductive patterns are common and present in several other societies [23, 30, 31]. Secondly, we aim to explore whether climatic variables, such as temperature and rainfall, might be associated with the reproductive pattern in the Baka. Several previous studies have found a link between weather variables and reproductive seasonality, in particular conception in periods of positive energy balance due to greater food availability and quality and thus better ovarian function [e.g. 16, 20, 27, 29, 30, 37]. If this is true, we would expect a conception peak in the Baka Pygmies in the dry season when the overall availability of wild food is greater [i.e. 59, 68]. Thirdly, other studies [22, 29, 31] state that during wetter seasons it is more likely to have spontaneous abortions or diseases affecting conceptions, so we would expect a conception peak during dry season. Fourthly, social factors might also play a role. Subsistence activities varies according to the seasons, and Baka men might spend days and weeks further from their families, engaged in hunting expeditions or agricultural related activities during the rainy seasons [56]. Also, as other studies have outlined high workload periods might decrease conception rates [21]. Thus, we expect conception rate to be lower during rainy seasons when subsistence activities away from the households are longer and the workload is higher. According to our previous hypotheses, we would expect a depression of fecundability during rainy seasons and a conception peak during dry seasons due to greater food availability, less disease incidence and subsistence activities closer to the households.. Finally, if climatic and reproductive patterns are related, our third aim is to investigate the relationship between global climate change and local reproductive patterns. Because temperatures are increasing worldwide and rainfall patterns are changing in different parts of the planet due to climate change, we expect to see variations in both variables over the last 30 years, with an effect on reproductive seasonality [e.g. 11, 27, 69].”

3. The description of the lifestyle. The Baka are called "hunter-gatherers" especially in the Abstract and Intro, and only later it is stated that they also practice some kind of subsistence agriculture, and hire out as labourers to other groups. This comes as a disappointment, after the strong claims about "hunter-gatherers". I think the complete lifestyle picture should be mentioned in the abstract, and the Intro / Methods.

We have moved the description of the Baka in the introduction, as suggested by the reviewer1, what we guess, allows to make clearer the lifestyle of the Baka at the early stage of the article. We have also added a sentence to better explain their livelihood. The Baka present a subsistence economy based on foraging activities, hunting, gathering and fishing, combined with a recently introduced small-scale farming (56) (lines 102-103).

Despite of their mixed economy, it seems to us that working with the Baka within the context of hunter-gatherers study is not in contradiction with their current livelihood which include small scale farming. We argue so because 1) many hunter-gatherers societies, or described as such are involved in other economic activities worldwide (see Codding and Kramer, 2016); 2) their main social and cultural livelihood of the Baka is still based on the fundational schemes of the hunter gatherer societies (egalitarism, sharing, allomaternal care, social organization) (Hewlett, 2014), and also their mobility and diet.

Codding, B.F., Kramer, K.L. (2016). Why forage? Hunters and gatherers in the twenty-first century. School for advanced research Press. University of New Mexico Press

Hewlett, B.S. 2014. Hunter-gatherer childhoods in the Congo Basin. In (B.S. Hewlett, ed.) Hunter-Gatherers of the Congo Basin: Culture, History, and Biology of African Pygmies, pp. 245–275. Transaction Publishers, New Brunswick.

4. Structuring causes for reproductive seasonality. Coming from mammals, the main cause is nutritional status and health status (this are factors basically common to all animals). In humans, there are additionally behavioural and socio-economic factors. It would be good to clearly structure these effects in the Intro, e.g. there are

(i) constraints on conception or pregnancy maintenance due to nutritional status, or infectious diseases (Malaria), or effects of climate on sperm quality

(ii) constraints on the frequency of mating / intercourse, due to behavioural effect of climate (discomfort), of workload (tiredness in certain seasons), of working patterns (e.g. working or hunting away from mates), cultural restrictions, and deliberate birth timing.

(iii) evolutionary ingrained triggers (e.g. phototriggers) that probably play a lesser role in humans in general, but also have little effect in the tropics.

In the population here, many causes for reduced conception/mating seem to co-incide in the wet seasons.

Thank you for this suggestions. Therefore, we chose to tackle this question in the Introduction by considering energetic status (nutrition), behavioral (socio-cultural-economic contexts) and climatic factors (see lines from 31 to 54).

5. Mentioning the magnitude of effect. Statistical significance is a necessary condition for the relevance of a finding, but more important is the magnitude of effect. Depending on how one calculates it, we get about 10% of all births in this population being due to seasonality, as opposed to about 90% being due to non-seasonal reproduction. This is never clearly stated in the manuscript, and is not part of the discussion about the reasons. But in my interpretation, stating this clearly would emphasize that while all the mentioned effects may have some effect, they are not overriding rulers of the Baka's reproduction. If you disagree about this conclusion, you should mention it and explain why it is wrong. Not mentioning the magnitude is just strange.

Also relates to comments:

 About line 137 - It would be good to specify what you consider a « seasonal pattern ». Your data suggests that a maximum of 15% or a more conservative estimate of 6% of all births are due to a seasonal peak (as opposed to a flatline throughout the year). This is not very much. This is probably not even enough to worry about seasonal effects. It is really important that you explain what magnitudes of effect you are expecting or you would consider relevant.

 In table 1 - if I use the “total” data, I get an average of 90 births per month. If I then quantify the number of births deviating from this, they are 99 in total. 99 of 1083 is 9%, i.e. the seasonal effect is responsible for 9% of all births. This is not very much. This is the biologically relevant magnitude that is more important than whether the pattern is significant. You need to display this and discuss this when talking about the relevance of seasonality in the reproductive life of the Baka.

 About line 278 – “The month of birth accounts for 8.4% of the variance in Dev.Birth” => This is an indication of the magnitude of the seasonality effect. I would prefer this being based on total number of births – but anyhow, this needs to be discussed in terms of how relevant the seasonal pattern is – and compared to other studies.

These are important comment raising many questions regarding the percentage of the variance explained by the explanatory variable, the magnitude of its effect and its relative importance as a phenomenon. 

About the variance explained by month of birth. Please note that the value that we are providing is the percentage of the variance in the monthly deviations to the yearly number of birth that is explained by the month of birth (8.4% in our analysis and mentioned in the part Results - Birth Seasonality). We now reiterate this statement in the discussion and discuss it with respect to sample size. Indeed, the percentage of variance explained by a variable depends on the sampling variance resulting from the granularity of the used categories. We chose to count births by months and years leading to little number of recorded births and large sample variance. Whether we would have count the number of birth each month over 10 years’ intervals would have increased sample size, decreased sampling variance, and increased the variance explained by month of birth (for instance up to 20% in this case). Another example is that of reviewer 1 taking the total number of birth by month for all years. In this case, obviously, month of birth explain 100% of the variance in number of birth per month. 

About the magnitude of the effect. This is an important and justified comment. For years 1980-1982, 1988-2002 the mean maximum deviations are +20% respectively for the first (in April) and second (in August) peaks and -20% (in January) and -38% (in June) respectively for the first and second minimums. For years 2003-2008, the first maximum (in April) is +38% and the second maximum (in October) is +29% while the first minimum (in January) is -18% and the second minimum (in August) is -48%. These deviations are statistically significant as tested by harmonic regression. These values are now given in lines 292-302.

It must be stressed that the magnitudes of these deviations is not small. For instance, at a whole other scale – that of country or ecological zone – maximum deviation observed by Dorélien (2016, table 3) range from +12% to +49%. (lines 302-304). Note that, in the harmonic regression, the magnitude of the effect is given by the magnitudes of cos and sin coefficients.

Of course, as rightfully emphasized by reviewer 1, the sum of these deviations concerns only about 9% of births, so that about 9% of the timing of birth is impacted by seasonality (but see following code for interpretational question about this value). This is however in the range of what has been observed elsewhere and – we think – far from being negligible. For instance, taking data in Ghana from Osei et al (2016) we calculate that 12% of timing of birth results from seasonality but with important impact for public health. 

Osei, E., Agbemefle, I., Kye-Duodu, G. et al. Linear trends and seasonality of births and perinatal outcomes in Upper East Region, Ghana from 2010 to 2014. BMC Pregnancy Childbirth 16, 48 (2016). https://doi.org/10.1186/s12884-016-0835-x

# Total number of births (all years) per month

 nbirth <- c(76,84, 84,101, 110,80, 88,81,92,105,93,89) 

 av <- mean(nbirth)

 perc.dev <- (nbirth-av)/av

 perc.dev

 plot(1:12, perc.dev, col="darkgrey", type="b", xlab="Month", ylab="Monthly deviations", xaxt="n", ylim=c(-50,+50))

 abline(h=0, col="red", lwd=1, lty=2)

 title("All years", cex.main =1)

 axis(1, labels=lab.month, at=1:12, cex.axis=0.7)

 # calculation suggested by reviewer 1 aiming to estimate the % of deviation from a uniform distribution

 sum(abs(tot-mean(tot)))/sum(tot)

 # please not that this metric strongly relates to standard deviation?

 sqrt(sum((tot-av)^2)/11)

 # Not that this metric can be large without resulting from seasonality (for instance if distribution is uniform)

 nbirth <- round(runif(12, min = 0, max = 200)) ; plot(nbirth, type="b")

 sum(abs(nbirth-mean(nbirth)))/sum(tot)

 # Or quite low while seasonality in specific month is really large 

 nbirth <- c(90,90,90,140,90,90,90,90,40,90,90,90); plot(nbirth, type="b")

 sum(abs(nbirth-mean(nbirth)))/sum(tot)

About Line 212 - You say that Dev.Birth is a percentage. The equation is not respresenting a percentage (one would need a « times hundred » for that, and one would need a denominator. I guess the equation is missing a « divided by » in front of the second Mj.

This is true and we thank a lot reviewer 1 to pin point this mistake. The right equation is:

〖"Dev.Birth" 〗_"ij" "=" (N_"ij" "-" "M" _"j" )/"M" _"j" ,

We have changed it and include the right equation in the manuscript, see Line 231.

About line 251 - I thought AIC already ‘punished’ for number of factors, so that an additional choice for models with few factors is not usually done ?

This is true. We should have specified: “the most parsimonious in terms of degree of freedom”. Please see lines 322-327. When several models exhibit a ΔAIC < 2 (so explaining similarly the data) it can be argued to show the one with the lowest number of degree of freedom. In the present case, because all explanatory variables are continuous variables, it means, for instance, that if two models A and A*B are equivalent in terms of AIC, we choose to discuss only the model A.

6. There are several specific instances where the text seems to contradict itself in short sequence, e.g. when saying that in the dry season, the diet is more diverse, but a few sentences later is is said that it is made up of 50% of one single kind of fruit. If it is dominated by a single fruit and YET more diverse, you shoud explain this so the reader does not feel confused. Another example is to say that the lack of such studies is "surprising" and then immediately explain why this data is hard to come by. If data are hard to come by, lack of work on it is not really a surprise .... Other instances indicated in the word file.

This is not contradictory. From the one side, there is a higher diversity of foods consumed during the dry season (many more different wild and crop species consumed) and at the same time, among this food, the bush mango, provided a large amount of nutritional intake. From the one side we are referring to the diversity of food, and from the other we are referring to the intake provided by some of these foods during the different dry seasons: the bush mango in the minor dry season (less intense dry season) and the wild meat and fish during the major dry season.

We have rephrase this part to make clearer the distinction (see lines 407-410). 

7. I would expect to have a clearer overview over the literature in terms of 'double peaks', e.g. Philibert et al. 2013 (I did not check other human literature).

Done.

About line 347-349 - I think you need to show the data (generally, one should also show non-significant results).

This is true. We now incorporate an additional figure in supplementary material (S3 Fig.) showing the absence of temporal change in the first minimum and maximum within the year.

8. The finding about an effect of global warming on the reproductive seasonality says that there is an effect on the second birht peak in the year (which occurs at later dates) but not on the first. This clearly means that the distance between these two peaks is increasing (or, in other words, the distance between the second peak and the subsequent first one of the next year is decreasing). This is not mentioned, and not analyzed.

Thank you for this suggestion, we mention it now in the lines 347-348, as followed “Thus, the distance between the two peaks tends to increase, and subsequently, the second peak and the subsequent first one of the next year is decreasing”. We are however not sure to understand what further analysis reviewer 1 would like us to performed. A linear regression on the difference of time between maximum 1 and 2 of the year demonstrates a significant increase (p=0.014) of this difference of about two days per year (in month beta=0.07388*30).

In particular, this should affect discussions about the "contstant" interbirth interval and the sentence that "siblings are likely to be borne at subsequent peaks". For example, one could correlate the dates of the peaks to test whether their distance increases, or correlate that distance with the years. Maybe in years where the second peak is particularly late, the next first peak of the following year is also delayed?

There is not a ‘constant’ interbirth interval, there is a modal IBI of 2.5 years. We added a comparison of IBI in two periods in lines 367-372. It reads “The modal 2.5-year IBI for the Baka is equivalent to five 6-month cycles. This means that successive sibling births will occur alternately in one peak or the other. The shift over time in the second peak would probably accompanies a change in the IBI. IBI data covers a shorter period than the records of birth, thus possible changes in the IBI over time are more difficult to assess. However, Ramirez Rozzi (58) has observed that the average IBI for mothers 15-25 years of age is 2.36 years in 2010 whereas it is 2.56 years in 2014. This change is not significant (P=0.384) [58].”

9. Very generally, the manuscript makes it clear that you do distinguish between birth and conception. Nevertheless, the choice of words often is as if you claimed that there is a direct effect of something, like temperature, on birthing. It is clear this is not how you mean it, yet the words sound like it. I recommend to mend this. One good way would be to replace the term "birth seasonality" by "reproductive seasonality" in many locations.

We appreciate this recommendation and we proceed to change ‘birth seasonality’ by ‘reproductive seasonality’. However, as the reviewer might also recognize, many authors use and analyses BIRTH seasonality. Therefore, we reproduce this term “birth seasonality” when rephrasing previous works, especially in our introduction.

About “I think wrong y-axis label in Fig. S2”

It was corrected

Reviewer 2

First of all, we want to thank very much reviewer Ed Hagen for its very careful and thorough review. It is an honor to be reviewed by someone taking the time to reproduce the analyses.

One this said, Pr. Hagen first asks a very interesting question: The Aka have a caterpillar season, which brings in substantial food and cash. Do the Baka have the same? 

There is also a caterpillar season in the area where the Baka live, that begins during the minor dry season (between June to august) but that is the most intense during the major rainy season (from August to December). 

However, the caterpillars are not usually sold by the Baka, but rather eaten directly by the families within the community. In that sense, there is no substantial income coming from the sale of caterpillars. In term of food intake, nutritional study would be needed to provide with an accurate answer. However, the period in which caterpillars are frequent also corresponds to the season in which the Baka use to eat less frequently meat from wildlife. Thus, while the amount of food provided by the caterpillars needs further research, we can say that the caterpillars might contribute to the overall dietary diversity of the Baka during this season.

Then, Pr. Hagen address several more statistical questions that we address below.

1. It would be nice if the authors provided an Excel spreadsheet in their SI.

Agreed. This is done in the revised submission.

2. The raw data are quite noisy, and a bimodal pattern is not readily apparent. Totalling births by month across all years, however (the bottom row of the authors’ Table 1), reveals a strong bimodal pattern.

We are happy that reviewer 2 emphasizes our concern about data granularity (which is a general one is statistics): aggregating data may reveal trends to the detriment of finer scale variations, while analyzing finer scale data increases the noises potentially concealing broader patterns. Often, data analyses starts with finding categories or procedures allowing the right balance between these two extremes. Here we could for instance have gathered births by five years intervals (actually we first did this and it gives similar results). Instead we chose to smooth data over 3 years intervals, a classical procedure in time series analysis.

3. Interestingly, a naive power spectrum analysis reveals a strong peak at 4 cycles per year. The authors used harmonic regression to test for either 1 peak or 2 peaks a year. This result suggests that there might instead be 4 peaks.

This is a very interesting thought. We must admit that we thin about testing for a third or a forth peak within the year. The reason is that climatologists have demonstrated a latitudinal pattern of seasonal precipitation in Africa: “Much of Africa is characterized by a single summer wet season, with a well-defined onset and end, during whichmost precipitation falls. Exceptions to the single wet season regime occur mostly near the equator, where two wet periods are usually separated by a period of relatively modest precipitation” (Liebmanne et al. 2012). Furthermore, to our knowledge, seasonality of birth with more than two peaks have never been documented.

This said, it is therefore especially clever to test for more peaks, a pattern that could very well occur due to cultural reasons or ‘hidden’ ecological ones. We therefore now test – still using harmonic regression – additional models with 3 and 4 peaks per year. However, whatever the methods used to compare models (Anova, AIC) the bimodal model is still the one fitting the better the smoothed DevBirth variable. This now indicated in the methods part Seasonal fluctuation in births.

Example – Comparing 2 vs 4 peaks using smoothed DevBirth for year 2003 and after.

# Model 2 peaks - The 6 below is here to test for complete cycle over 6 months, so 2 cycles per years

Time <- 1:length(Dev.After2003)

xc <-cos(2*pi*Time/6) 

xs <-sin(2*pi*Time/6)

fit.lm2 <- lm(Dev.After2003~xc+xs)

extractAIC(fit.lm2)

[1] 3.000 1497.813

# Model 4 peaks - The 3 below is here to test for complete cycle over 6 months, so 4 cycles per years

Time <- 1:length(Dev.After2003)

xc<-cos(2*pi*Time/4) 

xs<-sin(2*pi*Time/4)

fit.lm4 <- lm(Dev.After2003~xc+xs)

extractAIC(fit.lm4)

[1] 3.000 1518.611

So how to explain reviewer 2’s results ? First we think that reviewer 2 made a mistake by entering d2$Time, instead of d2$Births in its ‘R’ code. By doing this, reviewer 2 analyses the ‘periodicity’ of time not of the number of birth. When performing the spectral analysis on the following vector,

[1] 2001.000 2001.083 2001.167 2001.250 2001.333 2001.417 2001.500 2001.583 2001.667 […] 2018.500 2018.583 2018.667 2018.750 2018.833 2018.917.

we do also find the follwing graph :

I am not 100% sure to understand why we find a periodicity of 4 in this case, but I think it is due to the rounding of the digits.

To follow with the example above, when we plot the density of frequencies of the Dev.Birth variable for years after 2003

m2 <- spectrum(Dev.After2003 , log="no", plot=F)

plot(m2$freq*12, m2$spec*2, xlab="Frequency", ylab = 'Spectral density', type="l")

We do have a frequency of two periods per year (the one at 6 being a peak every two months)

 Liebmann, B., Bladé, I., Kiladis, G. N., Carvalho, L. M. V., B. Senay, G., Allured, D., Leroux, S., & Funk, C. (2012). Seasonality of African Precipitation from 1996 to 2009, Journal of Climate, 25(12), 4304-4322

Journal Requirements:

Done

2. Please complete the following questions designed to promote ethical global research practices. The information requested in this section should also be included in the Methods section of your manuscript. 

Did you obtain permissions from a local agency (e.g. government office, ethics committee or similar) to conduct this study? 

Provide details as to who granted local permissions and/or consent. Refer to any individuals involved by their role or title but do not list their name(s).

Did you obtain written informed consent from a representative of the local community or region before the research took place?

Did the individuals who provided consent for this research also agree to the sharing of the data obtained, as per your Data Availability Statement? 

How were the aims of the research investigation, its methodology, and its anticipated outcome(s) discussed with, and agreed upon by the people/s being studied or representatives of the local community? How did you establish who speaks for the community?

What aspects of the research process, including the right to exercise control over how the information and/or materials are collected, studied, and/or published, did the local community being studied have control over? 

How were the ethical values of the local community acknowledged, discussed, and/or incorporated into the research design?

Have the findings of the research been presented or made available in an understandable format to stakeholders in the community where the study was conducted (e.g. via a presentation, summary report, copies of publications, etc.)?

We have already specified all this points. First of all,, we are not working on living organisms but on archives (records of birth). Nevertheless, our work was carried out under international agreement and approved by the health committee of Cameroun. The full paragraph in the ms is as follows (lines 193-200) “Prior informed consent was obtained from all participants and the study was carried out with the approval of the chief of the Baka community. All methods are non-invasive and were carried out in accordance with relevant guidelines and regulations. This study have obtained approval from the Centre National de la Recherche Scientifique, the Agence National de la Recherche (France) and from the Institut de Recherche et Développement. It was carried out under an international agreement between the Institut de Recherche pour le Développement (IRD) and the Ministry of Scientific Research and Technology of Cameroon. It obtained the approval of the National Committee of Ethics for Research for Human Health of Cameroon (2018/06/1049/CE/CNERSH/SP).”

3. We note that you have referenced (ie. Bewick et al. [5]) which has currently not yet been accepted for publication. Please remove this from your References and amend this to state in the body of your manuscript: (ie “Bewick et al. [Unpublished]”) as detailed online in our guide for authors

We have never made reference to Bedwick et al’s work.

---

## [Decision Letter · Decision Letter 1]

25 Jan 2022

PONE-D-21-22413R1Reproductive Seasonality in the Baka Pygmies, environmental factors and climatic changesPLOS ONE

Dear Dr. Ramirez Rozzi,

Thank you for submitting your manuscript to PLOS ONE. After careful consideration, we feel that it has merit but does not fully meet PLOS ONE’s publication criteria as it currently stands. Therefore, we invite you to submit a revised version of the manuscript that addresses the points raised during the review process.

ACADEMIC EDITOR: This is an well done study that now meets Plos One criteria for publication. Please consider my decision of minor revision as an opportunity to finalize any small editorial changes. We do not have copy editing services, and I noticed a couple typos. Thank you for submitting your work to Plos One.

We look forward to receiving your revised manuscript.

Kind regards,

Jacob Freeman

Academic Editor

PLOS ONE

Journal Requirements:

Reviewers' comments:

Reviewer's Responses to Questions

**Comments to the Author**

1. If the authors have adequately addressed your comments raised in a previous round of review and you feel that this manuscript is now acceptable for publication, you may indicate that here to bypass the “Comments to the Author” section, enter your conflict of interest statement in the “Confidential to Editor” section, and submit your "Accept" recommendation.

Reviewer #2: All comments have been addressed

2. Is the manuscript technically sound, and do the data support the conclusions?

Reviewer #2: Yes

3. Has the statistical analysis been performed appropriately and rigorously? 

Reviewer #2: I Don't Know

4. Have the authors made all data underlying the findings in their manuscript fully available?

Reviewer #2: Yes

5. Is the manuscript presented in an intelligible fashion and written in standard English?

Reviewer #2: Yes

6. Review Comments to the Author

Reviewer #2: The authors carefully addressed all my comments. As the authors correctly point out, my one major concern was due to a coding error on my part.

Minor corrections:

line 100: "The rainy seasons are characterized by short periods of torrential rainfall"

line 156: "Subsistence activities vary according to the seasons, ..."

line 194: "This study has obtained approval from..."

line 360: "This study brought several findings: ..." is awkward. Maybe "This study had several results: ..." or similar.

7. PLOS authors have the option to publish the peer review history of their article (what does this mean?). If published, this will include your full peer review and any attached files.

Reviewer #2: No

---

## [Author Response · Author response to Decision Letter 1]

6 Feb 2022

Thank you to the reviewer and the Academic Editor for their comments.

There is no change in the financial disclosure.

Laboratory protocols are not applicable to this study.

ACADEMIC EDITOR: This is an well done study that now meets Plos One criteria for publication. Please consider my decision of minor revision as an opportunity to finalize any small editorial changes. We do not have copy editing services, and I noticed a couple typos.

We have realized the editorial (typo) changes (lines 13, 14, 28, 37, 42, 45, 68, 78, 79, etc.)

Journal Requirements:

References

References were checked and no change was needed.

Reviewers' comments:

The reviewer suggested some minor corrections, the following

Minor corrections:

line 100: "The rainy seasons are characterized by short periods of torrential rainfall"

line 156: "Subsistence activities vary according to the seasons, ..."

line 194: "This study has obtained approval from..."

line 360: "This study brought several findings: ..." is awkward. Maybe "This study had several results: ..." or similar.

Of course, these suggestions were all accepted and can be seen in the maked-up-copy of the ms.

The figures are now PACE generated ones.

---

## [Editor Report · Decision Letter 2]

17 Feb 2022

Reproductive Seasonality in the Baka Pygmies, environmental factors and climatic changes

PONE-D-21-22413R2

Dear Dr. Ramirez Rozzi,

We’re pleased to inform you that your manuscript has been judged scientifically suitable for publication and will be formally accepted for publication once it meets all outstanding technical requirements.

Kind regards,

Jacob Freeman

Academic Editor

PLOS ONE
---

## [Editor Report · Acceptance letter]

23 Feb 2022

PONE-D-21-22413R2 

Reproductive Seasonality in the Baka Pygmies, environmental factors and climatic changes 

Dear Dr. Ramirez Rozzi:

I'm pleased to inform you that your manuscript has been deemed suitable for publication in PLOS ONE. Congratulations! Your manuscript is now with our production department. 

Kind regards, 

on behalf of

Dr. Jacob Freeman 

Academic Editor

PLOS ONE